# cPAPERS: A Dataset of Situated and Multimodal Interactive Conversations in Scientific Papers

**Anirudh Sundar**[*]    **Jin Xu**[*]    **William Gay**[*]    **Christopher Richardson**[*]    **Larry Heck**

AI Virtual Assistant Lab
Georgia Institute of Technology
[*] Equal Contribution
{asundar34, jxu81, wgay7, crichardson8, larryheck}@gatech.edu

## Abstract

An emerging area of research in situated and multimodal interactive conversations (SIMMC) includes interactions in scientific papers. Since scientific papers are primarily composed of text, equations, figures, and tables, SIMMC methods must be developed specifically for each component to support the depth of inquiry and interactions required by research scientists. This work introduces CONVERSATIONAL PAPERS (cPAPERS), a dataset of conversational question-answer pairs from reviews of academic papers grounded in these paper components and their associated references from scientific documents available on arXiv. We present a data collection strategy to collect these question-answer pairs from OpenReview and associate them with contextual information from LaTeX source files. Additionally, we present a series of baseline approaches utilizing Large Language Models (LLMs) in both zero-shot and fine-tuned configurations to address the cPAPERS dataset.

## 1 Introduction

Developing conversational assistants capable of situated and multimodal interactive conversations (SIMMC) over structured knowledge sources remains an open problem [33, 47, 27].

An emerging area of research within this domain is conversational interactions over scientific documents [41]. The number of scientific articles has grown dramatically over the past decade [1], making it difficult for scientists to find, read, understand, and connect advancements published by their fellow researchers. Scientists in the field of biomedicine, for example, publish over 1 million articles per year or a new article every 2 minutes on average [28]. Developing methods to understand scientific documents and assist researchers is an important problem, especially for the Natural Language Processing (NLP) community. Scientific documents present an interesting challenge in SIMMC-based methods since the content is frequently multimodal [6].

Besides textual paragraphs, researchers rely on various modalities to describe research methods. Figures convey information about concepts developed throughout the paper. Generally, figures represent many different types of information including most commonly graphical and image information. For example, figures explain model architectures and pipelines, summarize experimental results in graphical plots, and represent complex information such as gradient learning surfaces and feature maps. In addition, figures can be used to show various stages of image processing in computer vision and image generation papers.

Equations are crucial for grasping mathematical concepts in scientific texts but can be challenging to interpret. Equations frequently rely on notation introduced elsewhere in the document, often requiring readers to draw from the entire paper to understand the formulation. In addition, equations

38th Conference on Neural Information Processing Systems (NeurIPS 2024) Track on Datasets and Benchmarks.

are typically related to and in many cases help the reader understand other paper components such as figures and tables (and vice-versa).

Finally, tables typically have structures that convey semantic information. These include the text in the row and column headings, and text and often links in the table cells. These structures summarize multiple concepts (e.g., experimental results) in a human-readable form. Therefore, developing SIMMC methods for scientific papers must include a semantic understanding of these structures and table content.

To advance the development of conversational assistants capable of SIMMC in scientific papers, this paper introduces Conversational Papers (cPAPERS[1]), a dataset of conversations in English situated in equations (cPAPERS-EQNS), figures (cPAPERS-FIGS), and tabular information (cPAPERS-TBLS) for scientific texts. Question-answer pairs are sourced from reviews and rebuttals from OpenReview.[2] Textual grounding to answer the questions and answers are sourced from the corresponding scientific articles hosted on arXiv [3], an open-access repository of academic preprints.

The contributions of this work include:

- Introduction of the CONVERSATIONAL PAPERS (cPAPERS) dataset including three splits (EQNS, FIGS, TBLS)
- Development of a novel and scalable approach for collecting question-answer pairs from OpenReview and linking them with relevant contextual information from open access TEX sources on arXiv
- Creation of baseline LLM approaches that utilize weakly grounded multimodal context for dialogue responses

The rest of the paper is organized as follows: Section 2 discusses related work in language modeling and question-answering for equations, tables, and figures. Section 3 provides a description of cPAPERS and details the dataset collection process. Next, Section 4 describes baseline approaches to address this dataset with a series of experiments utilizing zero-shot prompting and parameter-efficient fine-tuning. Section 5 discusses the results, while Sections 6 and 7 address the conclusion and limitations, respectively.

## 2   Related Work

### 2.1   Language Modeling for Equations

The challenge of modeling mathematical equations has become an active research area in Natural Language Processing. [11] proposes an encoder-decoder approach to generate equations from math word problems. [53] presents an approach for the dual problem of generating math word problems consistent with equations. Prior work has also proposed learning representations from equations using the Transformer encoder [39], Tree-based encoders [54], and RNNs [56]. More recent work utilizing generative architectures includes MathGPT [44], an auto-regressive model based on GPT-2 [40] for various language+equation tasks, and Nougat [6], a Visual Transformer to parse academic documents to markup language. In contrast, cPAPERS is a dataset that addresses both grounded conversational question-answering and the modeling of mathematical equations. Answering the questions accurately requires understanding the expressions in the equations while also formulating an accurate response that addresses the task succinctly.

### 2.2   Question-Answering over Figures

Early datasets on question-answering over figures include VQA [2], Visual7W [63], VisDial [13], MANYMODALQA [21], and OK-VQA [32, 42]. More recent datasets addressing the problem of open-domain conversations include IGC [34], MOD [17], and Image-Chat [45]. However, the images

---

[1]The dataset is publicly available at `https://huggingface.co/datasets/avalab/cPAPERS`. The collection process, baseline models, and code are located at `https://github.com/avalab-gt/cPAPERS` under the GNU General Public License.

[2]`https://openreview.net/`

[3]`https://arxiv.org`

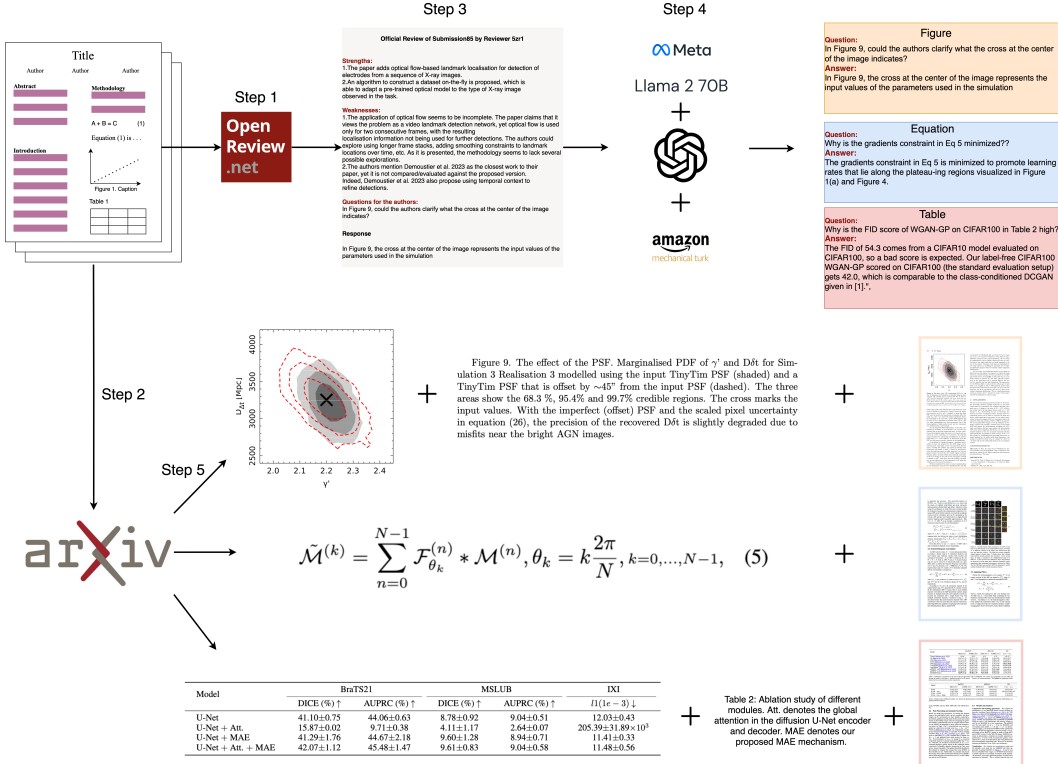

Figure 1: Overview of the data collection approach. First, the reviews are obtained from OpenReview. The corresponding TEX source is obtained from arXiv (only if the paper is also posted to arXiv). The reviews and rebuttals are filtered using a combination of LLMs and Amazon Mechanical Turk to obtain question-answer pairs in the json format. The TEX source is parsed to obtain the associated equation, table, or figure to create a dataset of questions and answers paired with the corresponding LATEX.

in these datasets are collected from MS-COCO [30] or YFCC100M [51], where visual content targets commonly seen everyday objects as opposed to scientific documents where images target specific information relevant to the explanation of a concept.

Recent work has addressed some of the challenges associated with multimodal image+text tasks situated in scientific documents. [50] presents a dataset of charts from scientific papers and associated natural language captions summarizing the information present in the chart. [19] develops a method to link labels with relevant images in patents. [12] introduces a dataset of 150 computer science papers, ground truth labels for the locations of figures, tables, and captions, and an approach to automatically extract this information from PDFs. SCICAP [25] is a much larger dataset of 400,000 figures, their captions, and associated textual references from various scientific papers. The dataset is collected by scraping scientific preprints from arXiv. For each figure, they provide the associated caption and all paragraphs that mention the figure.

While these datasets address tasks related to multimodal content in scientific documents, they are not conversational in nature. In contrast, cPAPERS addresses the shortcomings of prior datasets as a multimodal, conversational dataset grounded in scientific documents. The questions and answers address specific contextual visual information grounded in scientific documents.

## 2.3 Tabular Question-Answering

Tabular question-answering addresses the problem of extractive QA grounded in the information contained in specific cells of a table. Prior tabular QA datasets include those collected from Wikipedia, such as WIKITABLEQUESTIONS [38], ManyModalQA [22], TABERT [57], NQ-Tables [23], FEVEROUS [4], FeTaQA [36], HYBRIDIALOGUE [35], and HiTab [10]. Other tabular datasets

| | Equation | | | Table | | | Figure | | |
|---|---|---|---|---|---|---|---|---|---|
| | train | dev | test | train | dev | test | train | dev | test |
| # Unique Papers | 672 | 286 | 335 | 715 | 285 | 302 | 761 | 275 | 313 |
| # QA Pairs | 993 | 336 | 394 | 932 | 327 | 342 | 1052 | 297 | 357 |
| # Tokens (average) | | | | | | | | | |
| | train | dev | test | train | dev | test | train | dev | test |
| Question | 25 | 25 | 25 | 24 | 22 | 26 | 23 | 23 | 24 |
| Answer | 92 | 102 | 90 | 86 | 79 | 81 | 83 | 88 | 81 |
| Contexts | 10,232 | 11,851 | 12,288 | 2,981 | 2,746 | 2,610 | 433 | 400 | 431 |
| References | 7,323 | 8,413 | 9,517 | 1,757 | 1,645 | 1,427 | 366 | 375 | 323 |
| Neighboring Contexts | 1,144 | 1,153 | 1,084 | 994 | 1,043 | 925 | - | - | - |
| Neighboring References | 1,000 | 947 | 1,152 | 736 | 657 | 588 | - | - | - |

Table 1: Dataset Statistics

are constructed from financial reports - TAT-QA [62], FINQA [9], MULTIHIERTT [61], or arXiv - iTBLS [48].

Proposed approaches to address the tabular QA task include architectures based off of the Transformer encoder [57, 24, 8, 16, 31, 20, 55], decoder [18, 3, 58, 26, 59, 46], or both (encoder-decoder) [35, 14, 49, 43, 48].

In contrast to prior tabular datasets, cPAPERS-TBLS presents a new source of grounded questions and answers and to the best of our knowledge, is the first to be situated in tables from OpenReview. Additionally, the question-answer pairs are not factoids. Rather, they are conversational in nature.

# 3 The cPAPERS Dataset

## 3.1 Dataset Description

Conversational Papers (cPAPERS) is a dataset of conversations in English situated in scientific texts. cPAPERS consists of question-answer pairs pertaining to figures (cPAPERS-FIGS), equations (cPAPERS-EQNS), or tabular information (cPAPERS-TBLS) from scientific papers. These pairs are sourced from the official reviews and rebuttals of The Conference and Workshop on Neural Information Processing Systems (NeurIPS) and The International Conference on Learning Representations (ICLR) between 2020 and 2023, as available on OpenReview. cPAPERS comprises of 1723 question-answer pairs over equations, 1601 pairs over tables, and 1706 pairs over figures, totaling 5030 question-answer pairs spanning 2350 unique scientific papers. A detailed breakdown of the dataset is provided in Table 1.

Section 3.2 details the steps of collecting cPAPERS from official reviews on OpenReview, extracting question-answer pairs, and associating them with LaTeX source files from arXiv.

## 3.2 Dataset Collection

### 3.2.1 Step 1 - Extract Official Review and Comments

We leverage the OpenReview API to access official reviews and comments for each paper submission. These reviews, authored by conference reviewers, typically include a paper summary, strengths and weaknesses, specific questions, and limitations. Authors respond to these critiques in the comments, providing answers that address the reviewers' questions. This step involves downloading the official reviews and comments for all submissions.

### 3.2.2 Step 2 - Download LaTeX files

For each paper recorded in Step 1, we initiate a call to the arXiv API to verify the existence of the paper. If the paper is found, we proceed to download all associated LaTeX source files. If a paper cannot be located using the arXiv API, we exclude this submission from further processing.

### 3.2.3 Step 3 - Regex Filtering

As mentioned in Step 1, review-rebuttal pairs contain a summary of the paper and additional information that may not be related to equations, tables, or figures. Therefore, we use regular expressions to identify mentions of equations, tables, and figures within each review and comment pair. The following regular expressions are employed for this purpose:

```
Equation: re.compile(r'\b(?:Equation|Eq\.?)\s*\(?\s*\d+\)?\b',re.IGNORECASE)
```

```
Table: re.compile(r'\bTable\s*\d+|\btables\b',re.IGNORECASE)
```

```
Figure: re.compile(r'\b(?:Figure|fig)\.?\s*\d+\b', re.IGNORECASE)
```

These expressions efficiently determine whether equations, tables, and figures are mentioned in either the review or the rebuttal, resulting in a separate list of review-rebuttal pairs each containing explicit references to an equation, table, or figure in the paper. It is possible for a review to contain references to multiple modalities.

### 3.2.4 Step 4 - Question-Answer Extraction

To extract question-answer pairs from the regex-filtered reviews and rebuttals, an LLM that is instruction fine-tuned to align with human preferences is prompted *in-context* with a single example. For this process, we use LLAMA-2-70B-CHAT-HF [52].

The prompt is as follows:

'*Given this example Content, Question, and Answer, look through the Summary and Comments that are dedicated to the paper and extract Question and Answer pairs that specifically target a particular Figure/Equation/Table in the content. Questions may be under sections labeled similarly to "Weaknesses" or "Questions" or "Review" or "Clarity, Quality, Novelty And Reproducibility" in the comments, and Answers may be under sections labeled "Response to Reviewer" or "Rebuttal" or "Comments".*'

System Prompt: *"You are a helpful question answer extracting assistant. Our goal is to extract Question and Answer pairs pertaining to a specific a Figure/Equation/Table in the content. Your response should be in the format of [Question] <question> [Answer] <answer> Do not add any other unnecessary content in your response."*

**Post-processing + JSON reformatting.** GPT 3.5 [7] is used to reformat the question-answer pairs as JSON. The following system prompt is used.

Prompt: *You are a helpful assistant. Please find the Question-Answer pairs, and format it as a json ({"question_answers": [{ "question": "", "answer": "" },...]})*

For each question-answer pair, an additional regular expression is employed to extract the respective figure, equation, or table number following the keywords 'Figure', 'Fig', 'Equation', 'Eq', or 'Table'.

```
pattern = r'\d+'
numbers = re.findall(pattern, input_string)
```

### 3.2.5 Step 5 - Crowdworker Cleanup

A significant portion of academic reviews and rebuttals on OpenReview pertain to clarification questions as well as those related to fixing typos. While the LLM processing removes most of the questions and answers addresing typos, to further ensure the quality of the dataset, we employ crowd workers. The crowd workers (recruited using Amazon Mechanical Turk) ascertain whether a question-answer pair about an equation, table, or figure is technical in nature or not. For each question-answer pair, two crowd workers provide feedback and only those question-answer pairs with a consensus are retained. Additional details of the crowdworker setup is provided in Appendix A.5.

### 3.2.6 Step 6 - Contextualizing QA Pairs

In addition to extracting question-answer pairs from OpenReview, cPAPERS associates these question-answer pairs with referring text from the LaTeX source on arXiv. Often, the text surrounding figures,

equations, and tables in scientific documents provides additional relevant information. Regular expressions are employed to locate all instances of figures, equation, and tables in the LaTeX source using the `\begin{figure}`, `\begin{equation}`, and `\begin{table}` environments.

**Equation:** The equation is obtained by extracting content enclosed between `\begin{equation}` and `\end{equation}` tags. We provide all equations from the LaTeX source for a question-answer pair.

**Table:** The table is extracted from the content enclosed between `\begin{tabular}` and `\end{tabular}` tags. We provide all tables from the LaTeX source for a question-answer pair.

**Figure:** First, the content between `\begin{figure}` and `\end{figure}` tags is extracted. From this content, we obtain the caption (using `\caption`) and the figure path (using `\includegraphics`).

Additionally, for each modality, we provide context and references:

**Context:** The context represents the paragraph of text preceding and the paragraph of text following the equation, table, or figure referred to in the question-answer pair.

**References:** In addition, we report all referring text in a document associated with the equation, table, or figure in the question-answer pair. Within each environment, we search for a label (`\label`) and if it exists, we locate all references by searching for text using the label in LaTeX's `\ref` command.

Reviews of academic pre-prints provide a source for high-quality document-grounded questions and answers and success on the task requires contextual and multimodal understanding. However, one of the main challenges with matching questions and answers from OpenReview with their corresponding multimodal context from arXiv is that while reviewer and author references to equations, tables, and figures are purely ordinal (e.g. 'Equation 3', 'Figure 4', 'Table 2'), the LaTeX source does not provide any reliable approach to associate this ordinal position with the multimodal context in the final compiled document, specifically for equations and tables. Additionally, arXiv contains multiple versions of papers with no guarantees on consistency with the submission to OpenReview. As a result, the cPAPERS dataset reports both the position of the equation, table, or figure referred to in the question, and all equations and tables from that specific paper. Section 4 contains additional details on how this information is used in a baseline approach.

### 3.2.7 Step 7 - Additional Post-processing of Figures

Authors frequently utilize different graphical formats in their papers. To ensure consistency in our dataset, we convert all `.pdf` and `.eps` files to `.png` format using ImageMagick. [4]

## 4 Baseline Approaches

We experiment with zero-shot prompting and parameter-efficient fine-tuning for the baseline approaches. Zero-shot prompting involves querying pre-trained LLMs to answer the questions in the cPAPERS dataset without additional fine-tuning. In this approach, we use *off-the-shelf* Transformer decoder LLMs already pre-trained using a causal language modeling objective followed by supervised fine-tuning and reinforcement learning with human feedback to follow instructions. The LLM is not fine-tuned on cPAPERS and we do not provide examples in-context.

In addition, we experiment with parameter-efficient fine-tuning using QLoRA [15] to better align the model with the style of responses in the cPAPERS dataset. An LLM is fine-tuned independently for each modality (equation, table, or figure).

We serialize equations and tables using the corresponding LaTeX representations from the TeX source on arXiv. As previously indicated in Section 3.2.6, pairing questions and answers from OpenReview with LaTeX sources from arXiv presents a challenge with possible inconsistencies between the references in questions and answers to the corresponding sources in LaTeX. Therefore, we conduct question-answering experiments under two settings - utilizing all multimodal content, or utilizing a smaller subset of neighboring, weakly-grounded multimodal content. While using all multimodal content (equations or tables) ensures that the correct equation (or table) referred to by the question-answer pair is provided to the model, limitations on context length constrain the model from accurately

---

[4]https://imagemagick.org/

| Modality | Setting | ROUGE-1 | ROUGE-2 | ROUGE-L | METEOR | BERT | BLEU |
|---|---|---|---|---|---|---|---|
| Question (Q) | - | **0.194** | **0.065** | **0.144** | 0.240 | **0.825** | **0.038** |
| Q+Equation | Neighboring | 0.190 | 0.063 | 0.139 | **0.245** | 0.821 | 0.035 |
| | All | 0.170 | 0.056 | 0.123 | 0.218 | 0.740 | 0.035 |
| Q+Context | Neighboring | 0.186 | 0.063 | 0.137 | 0.237 | 0.809 | 0.036 |
| | All | 0.079 | 0.027 | 0.058 | 0.102 | 0.345 | 0.036 |
| Q+References | Neighboring | 0.176 | 0.061 | 0.129 | 0.223 | 0.764 | 0.036 |
| | All | 0.112 | 0.037 | 0.082 | 0.143 | 0.498 | 0.037 |

Table 2: Zero-shot language modeling on cPAPERS-EQNS test set with LLAMA-2-70B

using this information. Evidenced by prior work demonstrating the retrieval capabilities of generative LLMs when provided weakly-grounded context [49], we reconcile this situation by providing the model with a subset of equations or tables from the entire paper. First, we assign each equation or table with an ordinal position based on each instance of a \begin{equation} or \begin{table*} in the LATEX source. Then, supposing the question-answer pair refers to equation$_i$, the LLM is provided a subset of equations: {equation$_{i-1}$, equation$_i$, equation$_{i+1}$}. We utilize one-sided grounding in the situation where grounding from either boundary does not exist.

# 5 Results

The following sections contain results of the baseline approaches on the development and test splits. We report results from automatic evaluation - ROUGE [29] [5], METEOR [5] [6], BERTScore [7] [60], and BLEU [8] [37]. ROUGE, METEOR, and BLEU respectively measure the recall, F1 score, and precision of n-grams between model generated and reference text. BERTScore leverages contextual embeddings to measure the semantic similarity of model generated responses to the ground truth.

**Zero-shot.** First, we experiment with zero-shot language modeling with LLAMA-2-70B [52] to answer questions on cPAPERS-EQNS, cPAPERS-TBLS, and cPAPERS-FIGS. The results from Tables 2 and 3 indicate that utilizing the neighboring equations/tables provides a significant improvement over using all equations/tables to answer questions in the dataset, evidenced by up to 2x improvement in ROUGE, METEOR, and BERTScores. Additionally, the results from Table 2 indicate that utilizing the question alone without neighboring textual context (paragraphs of text surrounding the equation) results in best ROUGE and METEOR scores on cPAPERS-EQNS, whereas utilizing referring text yields the best scores on these metrics in cPAPERS-TBLS when compared to using the question alone to generate answers (Table 3). We posit this is because referring text in the document is predominantly used to summarize tabular results.

| Modality | Setting | ROUGE-1 | ROUGE-2 | ROUGE-L | METEOR | BERT | BLEU |
|---|---|---|---|---|---|---|---|
| Question (Q) | - | 0.192 | 0.058 | 0.136 | 0.232 | **0.832** | 0.028 |
| Q + Table | Neighboring | 0.206 | 0.061 | **0.145** | 0.237 | 0.828 | **0.031** |
| | All | 0.176 | 0.052 | 0.123 | 0.199 | 0.697 | 0.030 |
| Q + Context | Neighboring | 0.202 | 0.062 | 0.142 | 0.241 | 0.828 | **0.031** |
| | All | 0.160 | 0.050 | 0.114 | 0.195 | 0.666 | 0.030 |
| Q + References | Neighboring | **0.207** | **0.064** | 0.144 | **0.243** | 0.829 | **0.031** |
| | All | 0.186 | 0.057 | 0.130 | 0.223 | 0.758 | 0.030 |

Table 3: Zero-shot language modeling on cPAPERS-TBLS test set with LLAMA-2-70B

Table 4 contains results for zero-shot language modeling on cPAPERS-FIGS. In this situation, utilizing context performs the best across metrics.

---

[5]https://huggingface.co/spaces/evaluate-metric/rouge

[6]https://huggingface.co/spaces/evaluate-metric/meteor

[7]https://huggingface.co/spaces/evaluate-metric/bertscore

[8]https://huggingface.co/spaces/evaluate-metric/bleu

| Modality | ROUGE-1 | ROUGE-2 | ROUGE-L | METEOR | BERT | BLEU |
|---|---|---|---|---|---|---|
| Question (Q) | 0.185 | 0.065 | 0.137 | 0.238 | 0.833 | 0.036 |
| Q + Caption | 0.200 | 0.074 | 0.149 | 0.248 | 0.837 | 0.039 |
| Q + Context | **0.208** | **0.076** | **0.155** | **0.254** | **0.837** | **0.039** |
| Q + References | 0.205 | 0.075 | 0.154 | 0.251 | 0.837 | 0.038 |

Table 4: Zero-shot language modeling on cPAPERS-FIGS test set with LLAMA-2-70B

| Modality | ROUGE-1 | ROUGE-2 | ROUGE-L | METEOR | BERT | BLEU |
|---|---|---|---|---|---|---|
| Question (Q) | 0.309 | **0.138** | 0.248 | 0.215 | 0.860 | 0.062 |
| Q + Equation | **0.317** | 0.134 | **0.251** | **0.223** | **0.861** | 0.070 |
| Q + Context | 0.297 | 0.126 | 0.235 | 0.221 | 0.817 | 0.077 |
| Q + References | 0.283 | 0.122 | 0.224 | 0.217 | 0.777 | **0.081** |

Table 5: Results of fine-tuning LLAMA-2-7B on cPAPERS-EQNS test set

**Fine-tuning**. We also report results by fine-tuning LLAMA-2-7B using QLoRA [15]. Motivated by the zero-shot experiments, we only conduct fine-tuning using the *neighboring* setting for cPAPERS-EQNS and -TBLS.

Results of fine-tuning on cPAPERS-EQNS are detailed in Table 5. Utilizing the question with the equation outperforms the question alone or using the question along with referring text or the context on ROUGE-1 and ROUGE-L scores, METEOR, and BERTScore.

In contrast, the results from finetuning cPAPERS-TBLS and cPAPERS-FIGS in Tables 6 and 7 indicate that using just the question results in the best performance across most metrics. We believe that this is a result of the nature of questions and answers in reviews where authors often utilize tables and figures to supplement reviewer questions. This is supported by Table 8. In the cPAPERS-EQNS split of the dataset, 55.9% of questions refer to an equation while 77.6% of answers refer to an equation. On cPAPERS-TBLS and cPAPERS-FIGS, this difference is exacerbated, with answers referring to a table or a figure more often and questions referring to them less often when compared to equations.

| Modality | ROUGE-1 | ROUGE-2 | ROUGE-L | METEOR | BERT | BLEU |
|---|---|---|---|---|---|---|
| Question (Q) | **0.315** | **0.121** | **0.235** | 0.218 | **0.869** | 0.059 |
| Q + Table | 0.293 | 0.107 | 0.218 | 0.212 | 0.820 | 0.065 |
| Q + Context | 0.294 | 0.106 | 0.216 | **0.225** | 0.838 | **0.070** |
| Q + References | 0.292 | 0.106 | 0.214 | 0.218 | 0.816 | 0.068 |

Table 6: Results of fine-tuning LLAMA-2-7B on cPAPERS-TBLS test set

| Modality | ROUGE-1 | ROUGE-2 | ROUGE-L | METEOR | BERT | BLEU |
|---|---|---|---|---|---|---|
| Question (Q) | **0.329** | **0.155** | **0.269** | **0.250** | 0.859 | 0.083 |
| Q + Caption | 0.322 | 0.147 | 0.260 | 0.246 | **0.868** | 0.083 |
| Q + Context | 0.321 | 0.140 | 0.256 | **0.250** | 0.858 | **0.091** |
| Q + References | 0.311 | 0.131 | 0.244 | 0.247 | 0.843 | **0.091** |

Table 7: Results of fine-tuning LLAMA-2-7B on cPAPERS-FIGS test set

## 6    Conclusion

This paper introduces the CPAPERS dataset, which addresses the shortcomings of prior datasets as a multimodal, conversational dataset grounded in scientific documents. Leveraging reviews of academic papers grounded in equations, figures, and tables, as well as their associated references from scientific documents available on arXiv, this dataset aims to further advance the development of conversational

| Split | Question (%) | Answer (%) |
|---|---|---|
| cPAPERS-EQNS | 55.94 | 77.65 |
| cPAPERS-TBLS | 46.47 | 79.51 |
| cPAPERS-FIGS | 54.16 | 80.48 |

Table 8: Comparison between the percentage of questions that contain a reference to an Equation, Table, or Figure versus the percentage of answers

assistants capable of situated and multimodal interactive conversation within scientific papers. We conduct a series of experiments with zero-shot prompting to benchmark state-of-the-art models performance and perform several parameter-efficient fine-tuning for the baseline approaches. The experimental results provide valuable insights and create opportunities for enhancing and innovating the development of future conversational AI assistant.

## 7 Discussion

This paper introduces an innovative and scalable approach for collecting a large dataset of situated and multimodal interactive conversations related to scientific papers. This approach has the potential to facilitate the collection of extensive question-answer pairs across a wide range of scientific fields with ease. Collecting such a large and diverse dataset would benefit the research community by providing opportunities to develop more advanced AI-based research assistants for scientific research.

There may be concerns that creating an AI assistant to help automate the scientific discovery process could diminish the role of human scientists. While this may be a possible use of this dataset, our intention is to develop an AI research assistant that amplifies human scientists, empowering them to achieve greater scientific discoveries more efficiently.

**Limitations.** The key limitation of this dataset is the presence of mismatched figures, tables, or equations across different versions of the manuscripts. The *.tex* files of the paper on arXiv often undergo multiple revisions, and comments on Open Review are typically specific to a particular version. Authors frequently make changes in response to reviewer comments, which may involve additions, removals, or reordering of figures, equations, or tables. This presents a potential mismatch between the figures, equations, or tables and the question-and-answer pairs, thereby introducing additional challenges for language modeling.

## Acknowledgments and Disclosure of Funding

This work was supported by NSF IIS-2112633 and by CoCoSys, one of seven centers in JUMP 2.0, a Semiconductor Research Corporation (SRC) program sponsored by DARPA.

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

# A  Dataset Documentation

## A.1  Dataset Description

CPAPERS is a dataset of conversational question-answer pairs from reviews of academic papers grounded in these paper components and their associated references from scientific documents available on arXiv. For a detailed description and intended uses, please refer to 1.

## A.2  Dataset Accessibility

- The dataset is publicly available at `https://huggingface.co/datasets/avalab/cPAPERS`
- URL to Croissant metadata record for viewing and downloading by the reviewers: `https://huggingface.co/datasets/avalab/cPAPERS/blob/main/croissant.json`
- Code repository for collecting the dataset: `https://github.com/avalab-gt/cPAPERS`
- Code repository for reproducing the benchmark results: `https://github.com/avalab-gt/cPAPERS`

## A.3  Hosting, licensing, and maintenance plan

The dataset is licensed under Creative Commons (CC), and the code is licensed under the GNU General Public License (GPL). We plan to host and maintain this dataset on HuggingFace.

## A.4  Dataset Examples

Example question-answer pairs are provided in Tables 9 10 11, .

|  | Example |
|---|---|
| Question | "What does the symbol ˜ mean in Equation 1?" |
| Answer | "The symbol ˜ in Equation 1 represents "follows this distribution". It means that the probability distribution of the context C is defined as the distribution of the random variable  C." |
| Question | "Can you provide more information about what is meant by 'generative process in biological multi-agent trajectories' in L27 and L83?" |
| Answer | "The generative process refers to Eq. (2), which is a conceptual equation representing the generative process in animal behaviors." |
| Question | "How does the DeepMoD method differ from what is written in/after Eq 3?" |
| Answer | "We add noise only to $u$, with the percentage being of the standard deviation of the dataset. Adding noise to $x$ and $t$ has not been studied to our knowledge and falls out of the scope of this paper." |
| Question | "How to do the adaptive attack based on Eq.(16)? Maximizing the loss in Eq.(16)?" |
| Answer | "By Maximizing the loss in Eq (16) using an iterative method such as PGD on the end-to-end model we attempt to maximize the loss to cause misclassification while minimizing the regret to avoid detection." |
| Question | "How does the proposed method handle the imputed reward?" |
| Answer | "The proposed method uses the imputed reward in the second part of Equation 1, which corresponds to the empirical risk of the combined dataset." |

Table 9: Example QA Pairs in the cPAPERS-EQNS dataset

## A.5  Crowdworker Instructions

A significant portion of academic reviews and rebuttals pertain to clarification questions and fixing typos. While the LLM processing removes most of these spurious questions and answers, to further ensure the quality of the dataset we employ crowdworkers from Amazon Mechanical Turk to ascertain whether a question-answer pair about an equation, table, or figure is technical in nature or asks to fix

|  | Example |
|---|---|
| Question | "What is the purpose of Table 2 in the paper?" |
| Answer | "Table 2 is used to provide a comparison of the computational complexity of the proposed approach with state-of-the-art methods." |
| Question | "Optimal number of clusters affected by the number of classes or similarity between classes?" |
| Answer | "The authors have addressed this concern by including a new experiment in Table 4 of the revised paper's appendix. The result shows that the optimal number of clusters is less affected by the number of classes but more affected by the similarity between classes." |
| Question | "Can you clarify the values represented in Table 1?" |
| Answer | "The values in Table 1 represent the number of evasions, which shows the attack strength. Therefore, the higher the value, the stronger the attack." |
| Question | "The experiments in table 1 do not seem to favor the proposed method much; softmax is better or similar. Can the authors explain why this might be the case?" |
| Answer | "The proposed method reduces to empirical risk minimization with a proper loss, and the seemingly trivial solution $\phi=\Theta$ is often not optimal. This could explain why the proposed method might not perform as well as other methods in certain experiments. However, the authors hope that addressing concerns about the method's theoretical properties would be beneficial." |
| Question | "Does the first row of Table 2 correspond to the offline method?" |
| Answer | "Yes, the first row of Table 2 corresponds to the offline method." |

Table 10: Example QA Pairs in the cPAPERS-TBLS dataset

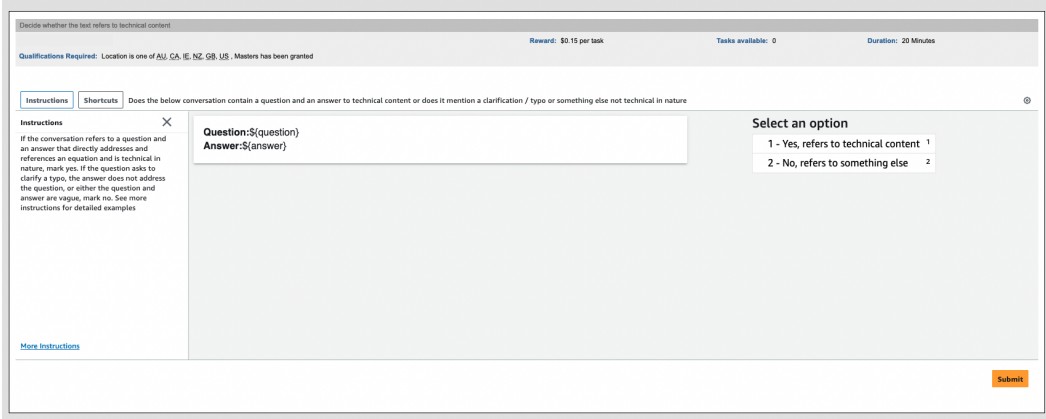

Figure 2: Screenshot of crowdworker interface on Amazon Mechanical Turk

a typo. We recruit crowdworkers with the masters qualification from predominantly English-speaking countries: Australia, Canada, Ireland, New Zealand, the United Kingdom, and the United States of America. Crowdworkers are compensated USD 0.15 per Human Intelligence Task, with each task taking an average of one minute to complete. The crowdworker instructions and interface are provided in Figure 2. The total cost of cleaning up the dataset including fees paid to Amazon Mechanical Turk is USD 3215. Since the dataset is collected from OpenReview, we did not identify any risks to crowdworkers.

| | Example |
|---|---|
| Question | "Why is there a gap between the proposed approach and the median approach in Fig. 5?" |
| Answer | "The gap is due to lower accuracy of the approximate median calculated by the bucketing scheme compared to a median method, albeit with faster epoch completion times." |
| Question | "What do experiments ensure us that the dependency is linear?" |
| Answer | "The linear dependency can be observed empirically in Figure 1. The paper will provide further experimental results in the updated version." |
| Question | "How do the different methods in Figure 5 have similar test errors, but the generalization bounds are so different?" |
| Answer | "The different methods in Figure 5 have similar test errors because they are all trained on the same dataset and have similar performance. However, the generalization bounds are different because they are computed using different methods. The proposed method uses a tighter bound that takes into account the structure of the ensemble, while the baselines use a looser bound that is based on an upper bound of the error rate." |
| Question | "What is the semantic meaning of "average episodic coverage" in Figure 5?" |
| Answer | "The semantic meaning of "average episodic coverage" in Figure 5 refers to the number of unique avatar positions. The authors have added a DIAYN baseline and a random agent baseline to provide context for how other methods fare." |
| Question | "In Figure 3, does the number of epochs mean the same thing for BAIL+ and MBAIL?" |
| Answer | "Yes, the number of epochs in Figure 3 means the same thing for BAIL+ and MBAIL. For MBAIL, it refers to the constant E in algorithm 2." |

Table 11: Example QA Pairs in the cPAPERS-FIGS dataset

# B  Language Modeling Details

We randomly split the dataset into training (60%), dev set (20%), and test set (20%). Hyperparameters for parameter-efficient fine-tuning can be found in 12. Other model training details are listed in 13.

| Parameter | Value |
|---|---|
| Rank | 64 |
| $\alpha$ | 16 |
| Dropout | 0.1 |

Table 12: Hyperparameters for Parameter-efficient fine-tuning using QLoRA

| Parameter | Value |
|---|---|
| Learning Rate | 2e-4 |
| Batch size | 4 |
| Warmup Schedule | Constant |
| Warmup Ratio | 0.03 |
| Epochs | 5 |
| Optimizer | `paged_adamw_32bit`[9] |
| Compute | 8 Nvidia A40 GPUs |

Table 13: Additional hyperparameters for fine-tuning experiments

## B.1  Additional Model Performance

We conducted additional experiments to benchmark the baseline performance of state-of-the-art pre-trained LLMs in answering questions in the cPAPERS dataset without additional fine-tuning. In Tables 14, 15, and 16, we report the zero-shot performance of LLAMA-2-7B, LLAMA-2-70B, LLAMA-3-8B, and LLAMA-3-70B on the cPAPERS dataset.

| Model | ROUGE-1 | ROUGE-2 | ROUGE-L | METEOR | BERTScore |
|---|---|---|---|---|---|
| LLAMA-2-7B | 0.189 | 0.060 | 0.136 | 0.232 | 0.823 |
| LLAMA-2-70B | 0.194 | 0.065 | 0.144 | 0.240 | 0.825 |
| LLAMA-3-8B | 0.139 | 0.047 | 0.107 | 0.161 | 0.768 |
| LLAMA-3-70B | 0.266 | 0.104 | 0.203 | 0.243 | 0.844 |

Table 14: Comparison of zero-shot performance across different models on the cPAPERS-EQNS test set

## B.2  Impact of Temperatures on Zero-shot Language Modeling Results

We conducted additional experiments to evaluate the influence of temperature on the baseline performance. Temperature parameters were sampled from 0.0 to 1.0 at intervals of 0.1, and each experiment was repeated five times with five randomly generated seeds. Average scores and standard errors across metrics were computed, and the results are presented in tables 17, 18, and 19.

## B.3  Impact of Temperatures on Fine-tuning Language Modeling Results

We conducted supplementary experiments to evaluate the influence of temperature on the fine-tuned model. Temperature parameters were sampled from 0.0 to 1.0 at intervals of 0.1, and each experiment was repeated five times with five randomly generated seeds. Average scores and standard errors across metrics were computed, and the results are presented in tables 20, 21, and 20.

| Model | ROUGE-1 | ROUGE-2 | ROUGE-L | METEOR | BERTScore |
|---|---|---|---|---|---|
| LLAMA-2-7B | 0.190 | 0.054 | 0.131 | 0.227 | 0.830 |
| LLAMA-2-70B | 0.192 | 0.058 | 0.136 | 0.232 | 0.832 |
| LLAMA-3-8B | 0.132 | 0.039 | 0.096 | 0.147 | 0.763 |
| LLAMA-3-70B | 0.256 | 0.086 | 0.187 | 0.217 | 0.850 |

Table 15: Comparison of zero-shot performance across different models on the cPAPERS-TBLS test set

| Model | ROUGE-1 | ROUGE-2 | ROUGE-L | METEOR | BERTScore |
|---|---|---|---|---|---|
| LLAMA-2-7B | 0.187 | 0.061 | 0.136 | 0.237 | 0.831 |
| LLAMA-2-70B | 0.185 | 0.065 | 0.137 | 0.238 | 0.833 |
| LLAMA-3-8B | 0.126 | 0.045 | 0.100 | 0.174 | 0.784 |
| LLAMA-3-70B | 0.282 | 0.119 | 0.218 | 0.256 | 0.853 |

Table 16: Comparison of zero-shot performance across different models on the cPAPERS-FIGS test set

| Temp | ROUGE-1 | ROUGE-2 | ROUGE-L | METEOR | BERTScore |
|---|---|---|---|---|---|
| 0.0 | $0.194 \pm 0.000$ | $0.065 \pm 0.000$ | $0.144 \pm 0.000$ | $0.240 \pm 0.000$ | $0.825 \pm 0.000$ |
| 0.1 | $0.193 \pm 0.001$ | $0.064 \pm 0.000$ | $0.143 \pm 0.000$ | $0.238 \pm 0.000$ | $0.825 \pm 0.000$ |
| 0.3 | $0.194 \pm 0.001$ | $0.065 \pm 0.001$ | $0.143 \pm 0.001$ | $0.240 \pm 0.000$ | $0.825 \pm 0.000$ |
| 0.5 | $0.194 \pm 0.001$ | $0.065 \pm 0.000$ | $0.142 \pm 0.000$ | $0.240 \pm 0.000$ | $0.825 \pm 0.000$ |
| 0.7 | $0.193 \pm 0.001$ | $0.065 \pm 0.001$ | $0.142 \pm 0.001$ | $0.239 \pm 0.000$ | $0.825 \pm 0.000$ |
| 0.9 | $0.193 \pm 0.001$ | $0.064 \pm 0.000$ | $0.141 \pm 0.001$ | $0.240 \pm 0.001$ | $0.825 \pm 0.000$ |

Table 17: Zero-shot performance (mean and standard errors over 5 seeds) of LLAMA-2-70B across different temperature on the cPAPERS-EQNS test set.

| Temp | ROUGE-1 | ROUGE-2 | ROUGE-L | METEOR | BERTScore |
|---|---|---|---|---|---|
| 0.0 | $0.192 \pm 0.000$ | $0.058 \pm 0.000$ | $0.136 \pm 0.000$ | $0.232 \pm 0.000$ | $0.832 \pm 0.000$ |
| 0.1 | $0.192 \pm 0.001$ | $0.058 \pm 0.000$ | $0.136 \pm 0.001$ | $0.231 \pm 0.000$ | $0.832 \pm 0.000$ |
| 0.3 | $0.191 \pm 0.001$ | $0.058 \pm 0.000$ | $0.136 \pm 0.000$ | $0.232 \pm 0.000$ | $0.832 \pm 0.000$ |
| 0.5 | $0.192 \pm 0.000$ | $0.059 \pm 0.000$ | $0.137 \pm 0.000$ | $0.233 \pm 0.001$ | $0.832 \pm 0.000$ |
| 0.7 | $0.192 \pm 0.001$ | $0.058 \pm 0.001$ | $0.137 \pm 0.001$ | $0.233 \pm 0.000$ | $0.832 \pm 0.000$ |
| 0.9 | $0.191 \pm 0.001$ | $0.057 \pm 0.001$ | $0.136 \pm 0.001$ | $0.232 \pm 0.000$ | $0.832 \pm 0.001$ |

Table 18: Zero-shot performance (mean and standard errors over 5 seeds) of LLAMA-2-70B across different temperature on the cPAPERS-TBLS test set.

| Temp | ROUGE-1 | ROUGE-2 | ROUGE-L | METEOR | BERTScore |
|---|---|---|---|---|---|
| 0.0 | $0.185 \pm 0.000$ | $0.065 \pm 0.000$ | $0.137 \pm 0.000$ | $0.238 \pm 0.000$ | $0.833 \pm 0.000$ |
| 0.1 | $0.188 \pm 0.001$ | $0.066 \pm 0.000$ | $0.140 \pm 0.000$ | $0.241 \pm 0.001$ | $0.834 \pm 0.000$ |
| 0.3 | $0.189 \pm 0.001$ | $0.067 \pm 0.001$ | $0.141 \pm 0.000$ | $0.241 \pm 0.001$ | $0.834 \pm 0.000$ |
| 0.5 | $0.191 \pm 0.001$ | $0.068 \pm 0.001$ | $0.143 \pm 0.001$ | $0.242 \pm 0.001$ | $0.835 \pm 0.000$ |
| 0.7 | $0.190 \pm 0.001$ | $0.067 \pm 0.000$ | $0.142 \pm 0.000$ | $0.241 \pm 0.001$ | $0.834 \pm 0.000$ |
| 0.9 | $0.190 \pm 0.001$ | $0.066 \pm 0.001$ | $0.141 \pm 0.000$ | $0.240 \pm 0.000$ | $0.834 \pm 0.000$ |

Table 19: Zero-shot performance (mean and standard errors over 5 seeds) of LLAMA-2-70B across different temperature on the cPAPERS-FIGS test set.

| Modality | Temp | ROUGE-1 | ROUGE-2 | ROUGE-L | METEOR | BERTScore |
|---|---|---|---|---|---|---|
| Question (Q) | 0.0 | $0.309 \pm 0.000$ | $0.138 \pm 0.000$ | $0.248 \pm 0.000$ | $0.215 \pm 0.000$ | $0.860 \pm 0.000$ |
| | 0.1 | $0.309 \pm 0.001$ | $0.139 \pm 0.001$ | $0.249 \pm 0.000$ | $0.216 \pm 0.000$ | $0.860 \pm 0.001$ |
| | 0.3 | $0.306 \pm 0.001$ | $0.135 \pm 0.000$ | $0.245 \pm 0.001$ | $0.213 \pm 0.000$ | $0.858 \pm 0.002$ |
| | 0.5 | $0.301 \pm 0.001$ | $0.132 \pm 0.001$ | $0.242 \pm 0.000$ | $0.212 \pm 0.001$ | $0.860 \pm 0.001$ |
| | 0.7 | $0.297 \pm 0.001$ | $0.125 \pm 0.001$ | $0.236 \pm 0.001$ | $0.210 \pm 0.001$ | $0.858 \pm 0.001$ |
| | 0.9 | $0.285 \pm 0.002$ | $0.113 \pm 0.001$ | $0.223 \pm 0.001$ | $0.202 \pm 0.002$ | $0.856 \pm 0.001$ |
| Q + Equation | 0.0 | $0.317 \pm 0.000$ | $0.134 \pm 0.000$ | $0.251 \pm 0.000$ | $0.223 \pm 0.000$ | $0.861 \pm 0.000$ |
| | 0.1 | $0.313 \pm 0.001$ | $0.133 \pm 0.001$ | $0.250 \pm 0.001$ | $0.220 \pm 0.001$ | $0.860 \pm 0.001$ |
| | 0.3 | $0.310 \pm 0.001$ | $0.130 \pm 0.001$ | $0.245 \pm 0.001$ | $0.221 \pm 0.001$ | $0.859 \pm 0.002$ |
| | 0.5 | $0.305 \pm 0.002$ | $0.124 \pm 0.001$ | $0.239 \pm 0.002$ | $0.217 \pm 0.002$ | $0.857 \pm 0.001$ |
| | 0.7 | $0.297 \pm 0.002$ | $0.116 \pm 0.001$ | $0.231 \pm 0.001$ | $0.213 \pm 0.002$ | $0.857 \pm 0.001$ |
| | 0.9 | $0.281 \pm 0.002$ | $0.102 \pm 0.002$ | $0.215 \pm 0.001$ | $0.206 \pm 0.002$ | $0.850 \pm 0.001$ |
| Q + Context | 0.0 | $0.297 \pm 0.000$ | $0.126 \pm 0.000$ | $0.235 \pm 0.000$ | $0.221 \pm 0.000$ | $0.817 \pm 0.000$ |
| | 0.1 | $0.297 \pm 0.001$ | $0.126 \pm 0.001$ | $0.235 \pm 0.000$ | $0.220 \pm 0.001$ | $0.820 \pm 0.001$ |
| | 0.3 | $0.297 \pm 0.002$ | $0.123 \pm 0.001$ | $0.234 \pm 0.001$ | $0.222 \pm 0.001$ | $0.824 \pm 0.001$ |
| | 0.5 | $0.297 \pm 0.002$ | $0.119 \pm 0.001$ | $0.232 \pm 0.002$ | $0.221 \pm 0.001$ | $0.832 \pm 0.002$ |
| | 0.7 | $0.287 \pm 0.001$ | $0.108 \pm 0.002$ | $0.219 \pm 0.003$ | $0.218 \pm 0.001$ | $0.832 \pm 0.002$ |
| | 0.9 | $0.270 \pm 0.002$ | $0.089 \pm 0.003$ | $0.200 \pm 0.004$ | $0.207 \pm 0.002$ | $0.826 \pm 0.001$ |
| Q + References | 0.0 | $0.283 \pm 0.000$ | $0.122 \pm 0.000$ | $0.224 \pm 0.000$ | $0.217 \pm 0.000$ | $0.777 \pm 0.000$ |
| | 0.1 | $0.286 \pm 0.001$ | $0.123 \pm 0.001$ | $0.226 \pm 0.001$ | $0.222 \pm 0.001$ | $0.785 \pm 0.002$ |
| | 0.3 | $0.283 \pm 0.001$ | $0.118 \pm 0.001$ | $0.222 \pm 0.001$ | $0.221 \pm 0.001$ | $0.786 \pm 0.003$ |
| | 0.5 | $0.281 \pm 0.002$ | $0.115 \pm 0.001$ | $0.219 \pm 0.001$ | $0.221 \pm 0.002$ | $0.785 \pm 0.003$ |
| | 0.7 | $0.271 \pm 0.001$ | $0.106 \pm 0.001$ | $0.208 \pm 0.002$ | $0.216 \pm 0.002$ | $0.781 \pm 0.002$ |
| | 0.9 | $0.258 \pm 0.001$ | $0.093 \pm 0.001$ | $0.193 \pm 0.002$ | $0.209 \pm 0.002$ | $0.770 \pm 0.002$ |

Table 20: Zero-shot performance (mean and standard errors over 5 seeds) of LLAMA-2-70B across different temperatures on the cPAPERS-EQNS test set.

| Modality | Temp | ROUGE-1 | ROUGE-2 | ROUGE-L | METEOR | BERTScore |
|---|---|---|---|---|---|---|
| Question (Q) | 0.0 | $0.315 \pm 0.000$ | $0.121 \pm 0.000$ | $0.235 \pm 0.000$ | $0.218 \pm 0.000$ | $0.869 \pm 0.000$ |
| | 0.1 | $0.316 \pm 0.001$ | $0.122 \pm 0.001$ | $0.234 \pm 0.000$ | $0.220 \pm 0.001$ | $0.869 \pm 0.001$ |
| | 0.3 | $0.313 \pm 0.001$ | $0.120 \pm 0.001$ | $0.232 \pm 0.001$ | $0.220 \pm 0.001$ | $0.869 \pm 0.001$ |
| | 0.5 | $0.309 \pm 0.001$ | $0.116 \pm 0.001$ | $0.229 \pm 0.001$ | $0.216 \pm 0.001$ | $0.868 \pm 0.000$ |
| | 0.7 | $0.302 \pm 0.002$ | $0.108 \pm 0.002$ | $0.221 \pm 0.001$ | $0.213 \pm 0.001$ | $0.868 \pm 0.001$ |
| | 0.9 | $0.288 \pm 0.003$ | $0.095 \pm 0.002$ | $0.208 \pm 0.002$ | $0.205 \pm 0.003$ | $0.865 \pm 0.000$ |
| Q + Table | 0.0 | $0.293 \pm 0.000$ | $0.107 \pm 0.000$ | $0.218 \pm 0.000$ | $0.212 \pm 0.000$ | $0.820 \pm 0.000$ |
| | 0.1 | $0.288 \pm 0.002$ | $0.105 \pm 0.001$ | $0.214 \pm 0.001$ | $0.211 \pm 0.001$ | $0.815 \pm 0.003$ |
| | 0.3 | $0.289 \pm 0.002$ | $0.103 \pm 0.001$ | $0.214 \pm 0.001$ | $0.215 \pm 0.002$ | $0.837 \pm 0.003$ |
| | 0.5 | $0.286 \pm 0.001$ | $0.096 \pm 0.001$ | $0.207 \pm 0.001$ | $0.216 \pm 0.001$ | $0.844 \pm 0.003$ |
| | 0.7 | $0.277 \pm 0.002$ | $0.085 \pm 0.001$ | $0.196 \pm 0.001$ | $0.212 \pm 0.004$ | $0.844 \pm 0.003$ |
| | 0.9 | $0.249 \pm 0.002$ | $0.064 \pm 0.001$ | $0.170 \pm 0.001$ | $0.200 \pm 0.004$ | $0.833 \pm 0.004$ |
| Q + Context | 0.0 | $0.294 \pm 0.000$ | $0.106 \pm 0.000$ | $0.216 \pm 0.000$ | $0.225 \pm 0.000$ | $0.838 \pm 0.000$ |
| | 0.1 | $0.301 \pm 0.001$ | $0.108 \pm 0.000$ | $0.219 \pm 0.000$ | $0.227 \pm 0.000$ | $0.846 \pm 0.002$ |
| | 0.3 | $0.301 \pm 0.001$ | $0.108 \pm 0.001$ | $0.218 \pm 0.001$ | $0.227 \pm 0.001$ | $0.846 \pm 0.001$ |
| | 0.5 | $0.297 \pm 0.002$ | $0.104 \pm 0.001$ | $0.214 \pm 0.002$ | $0.226 \pm 0.002$ | $0.848 \pm 0.002$ |
| | 0.7 | $0.287 \pm 0.002$ | $0.092 \pm 0.001$ | $0.203 \pm 0.002$ | $0.218 \pm 0.003$ | $0.845 \pm 0.002$ |
| | 0.9 | $0.272 \pm 0.002$ | $0.079 \pm 0.001$ | $0.187 \pm 0.002$ | $0.213 \pm 0.003$ | $0.842 \pm 0.001$ |
| Q + Reference | 0.0 | $0.292 \pm 0.000$ | $0.106 \pm 0.000$ | $0.214 \pm 0.000$ | $0.218 \pm 0.000$ | $0.816 \pm 0.000$ |
| | 0.1 | $0.297 \pm 0.001$ | $0.108 \pm 0.001$ | $0.217 \pm 0.001$ | $0.222 \pm 0.001$ | $0.830 \pm 0.002$ |
| | 0.3 | $0.298 \pm 0.002$ | $0.105 \pm 0.001$ | $0.217 \pm 0.001$ | $0.224 \pm 0.001$ | $0.842 \pm 0.002$ |
| | 0.5 | $0.294 \pm 0.002$ | $0.101 \pm 0.001$ | $0.212 \pm 0.002$ | $0.221 \pm 0.001$ | $0.843 \pm 0.003$ |
| | 0.7 | $0.288 \pm 0.001$ | $0.092 \pm 0.001$ | $0.205 \pm 0.001$ | $0.219 \pm 0.001$ | $0.847 \pm 0.001$ |
| | 0.9 | $0.272 \pm 0.002$ | $0.081 \pm 0.001$ | $0.190 \pm 0.001$ | $0.212 \pm 0.002$ | $0.837 \pm 0.002$ |

Table 21: Zero-shot performance (mean and standard errors over 5 seeds) of LLAMA-2-70B across different temperatures on the cPAPERS-TBLS test set.

| Modality | Temp | ROUGE-1 | ROUGE-2 | ROUGE-L | METEOR | BERTScore |
|---|---|---|---|---|---|---|
| Question (Q) | 0.0 | $0.329 \pm 0.000$ | $0.155 \pm 0.000$ | $0.269 \pm 0.000$ | $0.250 \pm 0.000$ | $0.859 \pm 0.000$ |
| | 0.1 | $0.333 \pm 0.001$ | $0.157 \pm 0.001$ | $0.272 \pm 0.001$ | $0.252 \pm 0.001$ | $0.859 \pm 0.002$ |
| | 0.3 | $0.329 \pm 0.002$ | $0.153 \pm 0.001$ | $0.267 \pm 0.001$ | $0.249 \pm 0.001$ | $0.860 \pm 0.003$ |
| | 0.5 | $0.324 \pm 0.002$ | $0.145 \pm 0.002$ | $0.259 \pm 0.002$ | $0.244 \pm 0.002$ | $0.861 \pm 0.002$ |
| | 0.7 | $0.315 \pm 0.002$ | $0.134 \pm 0.002$ | $0.248 \pm 0.002$ | $0.242 \pm 0.002$ | $0.862 \pm 0.001$ |
| | 0.9 | $0.291 \pm 0.001$ | $0.113 \pm 0.002$ | $0.223 \pm 0.001$ | $0.232 \pm 0.002$ | $0.857 \pm 0.003$ |
| Question + Figure | 0.0 | $0.322 \pm 0.000$ | $0.147 \pm 0.000$ | $0.260 \pm 0.000$ | $0.246 \pm 0.000$ | $0.868 \pm 0.000$ |
| | 0.1 | $0.325 \pm 0.001$ | $0.149 \pm 0.001$ | $0.262 \pm 0.001$ | $0.246 \pm 0.001$ | $0.865 \pm 0.001$ |
| | 0.3 | $0.322 \pm 0.002$ | $0.146 \pm 0.001$ | $0.258 \pm 0.002$ | $0.245 \pm 0.002$ | $0.865 \pm 0.001$ |
| | 0.5 | $0.318 \pm 0.001$ | $0.139 \pm 0.001$ | $0.251 \pm 0.001$ | $0.242 \pm 0.001$ | $0.865 \pm 0.001$ |
| | 0.7 | $0.306 \pm 0.001$ | $0.125 \pm 0.001$ | $0.239 \pm 0.001$ | $0.238 \pm 0.002$ | $0.863 \pm 0.001$ |
| | 0.9 | $0.281 \pm 0.001$ | $0.102 \pm 0.002$ | $0.212 \pm 0.001$ | $0.226 \pm 0.001$ | $0.852 \pm 0.003$ |
| Context | 0.0 | $0.321 \pm 0.000$ | $0.140 \pm 0.000$ | $0.256 \pm 0.000$ | $0.250 \pm 0.000$ | $0.858 \pm 0.000$ |
| | 0.1 | $0.317 \pm 0.002$ | $0.138 \pm 0.001$ | $0.254 \pm 0.001$ | $0.248 \pm 0.001$ | $0.853 \pm 0.002$ |
| | 0.3 | $0.323 \pm 0.002$ | $0.138 \pm 0.001$ | $0.255 \pm 0.001$ | $0.252 \pm 0.001$ | $0.861 \pm 0.001$ |
| | 0.5 | $0.318 \pm 0.001$ | $0.132 \pm 0.002$ | $0.248 \pm 0.002$ | $0.250 \pm 0.002$ | $0.863 \pm 0.001$ |
| | 0.7 | $0.307 \pm 0.001$ | $0.122 \pm 0.002$ | $0.236 \pm 0.002$ | $0.242 \pm 0.002$ | $0.861 \pm 0.001$ |
| | 0.9 | $0.278 \pm 0.001$ | $0.097 \pm 0.002$ | $0.209 \pm 0.002$ | $0.225 \pm 0.002$ | $0.851 \pm 0.001$ |
| Reference | 0.0 | $0.311 \pm 0.000$ | $0.131 \pm 0.000$ | $0.244 \pm 0.000$ | $0.247 \pm 0.000$ | $0.843 \pm 0.000$ |
| | 0.1 | $0.314 \pm 0.002$ | $0.134 \pm 0.001$ | $0.248 \pm 0.001$ | $0.250 \pm 0.001$ | $0.848 \pm 0.003$ |
| | 0.3 | $0.310 \pm 0.002$ | $0.130 \pm 0.001$ | $0.244 \pm 0.001$ | $0.249 \pm 0.002$ | $0.845 \pm 0.002$ |
| | 0.5 | $0.308 \pm 0.002$ | $0.125 \pm 0.001$ | $0.240 \pm 0.001$ | $0.249 \pm 0.002$ | $0.857 \pm 0.003$ |
| | 0.7 | $0.296 \pm 0.001$ | $0.113 \pm 0.001$ | $0.226 \pm 0.001$ | $0.241 \pm 0.001$ | $0.854 \pm 0.002$ |
| | 0.9 | $0.269 \pm 0.001$ | $0.090 \pm 0.001$ | $0.198 \pm 0.002$ | $0.226 \pm 0.002$ | $0.843 \pm 0.001$ |

Table 22: Zero-shot performance (mean and standard errors over 5 seeds) of LLAMA-2-70B across different temperatures on the cPAPERS-FIGS test set.

