# OpenReview forum: "cPAPERS: A Dataset of Situated and Multimodal Interactive Conversations in Scientific Papers"
_NeurIPS.cc/2024/Datasets_and_Benchmarks_Track — NeurIPS 2024 Track Datasets and Benchmarks Poster_

### Official Review · Reviewer_xcYG · 2024-07-24

**Rating:** 8
**Confidence:** 3

**Review:**

Overall, I think this paper makes a useful contribution to the community. I hope the authors will find the following notes helpful.

A few questions for the authors:
- Per line 168, it seems that "clarification questions" are being filtered out. Why is this the case? The motivating example in Figure 1, at the top, seems to be one. Could this (perhaps ironically) be clarified?
- Line 255 mentions an ANOVA, but what exactly is being reported here? I think it needs to be clear: (1) what exactly the independent and dependent variables are, (2) how the assumptions of the ANOVA (at least the important ones) were determined to be met, (3) what the results showed (F-statistic, degrees of freedom, exact p value), (4) if any post-hoc tests were conducted and how, etc. Otherwise, this statement is not clear.
- Line 246 appears to hypothesize that referring text for tables summarize the tabulated results, and hence is helpful for question answering. However, if this were the case, wouldn't it mean that the questions can be answered on the basis of the table alone, since the summary can simply be extracted from the table? I.e. the information is already present. Could this be clarified?
- What does "authors often utilize tables and figures to supplement reviewer questions" mean exactly? The authors produce the tables and figures before the reviewer questions arise, so how do the authors "supplement" the reviewer questions? I'm not sure I followed the point being made.


Minor notes:
- Line 219: "\LaTeXrepresentations" -> "$\LaTeX$ representations"?
- Similar issue on 221, likely needs a ~ after the \LaTeX command.

**Strengths:**

I think this work is likely to be useful to the community. It’s very clever to use questions and responses from OpenReview, since these are ecologically valid by default (vs. e.g. synthetically constructed questions). This also reflects a distribution likely rarely seen in datasets: the questions are information-seeking questions asked by experts, which means they are likely biased towards being more detailed or thought-out. The methodology is well developed, using a combination of the OpenReview API, LLM processing, and crowd work to assemble a resource that is challenging to otherwise obtain.

**Additional Feedback:**

N/A

**Clarity:**

Overall, the paper is clear (see some exceptions to this in my review notes). The paper is overall well constructed and easy to follow.

**Correctness:**

Generally, I think the work is well done. Some specific comments are given in my review.

**Documentation:**

Yes.

**Ethics:**

I don’t think this paper introduces any new ethical concerns.

**Limitations:**

Good consideration of limitations, but I would also encourage considering the following ones:
- Bias towards expert-level questions. In real-world QA contexts, the reader might not pay as much attention or have as much expertise as an assigned reviewer of a paper. So, toward the goal of developing SIMMC, there may be a distribution shift between the questions and answers in the paper and those that would be useful for many users in practice.
- Lack of comparison to simpler baselines. For example, extracting figures, tables, and equations, and then simply generating QA pairs directly from these (which has the advantages of virtually infinite scalability only bound by compute or cost, a promptable level of difficulty/expertise, etc.)
- It might be useful to consider what happens when a question addresses a figure and a table both, for example.
- Reviewer questions may not always be answerable from the content of the paper. It may require e.g. additional experiments, or knowledge not reported in the paper.

**Opportunities For Improvement:**

I’ve mentioned a few things in my review, but want to also note that an alternate strategy for this dataset is to use something like GROBID or Nougat (+ things like PDFFigures2) to extract representations of papers from the OpenReview source directly (to ensure they are matched to the comments). Just something to consider.

**Relation To Prior Work:**

Yes.

**Summary And Contributions:**

This paper contributes a dataset and initial benchmark for question-answering based on academic papers and real questions asked by reviewers, with a focus on tables, equations, and figures.

---

> ### Author Rebuttal · Authors · 2024-08-16
>
> Thank you for your comments! Here are some responses to the concerns raised:
>
> * Per line 168, it seems that "clarification questions" are being filtered out. Why is this the case? The motivating example in Figure 1, at the top, seems to be one. Could this (perhaps ironically) be clarified?
>   * Thanks for pointing this out, we are not filtering out clarification questions, but we see how it could be interpreted this way. We will rewrite this part of the paper to add an additional line saying “we wish to retain the clarification questions and other questions that are technical in nature while removing the spurious questions addressing typos”
>
> * Line 255 mentions an ANOVA, but what exactly is being reported here? I think it needs to be clear: (1) what exactly the independent and dependent variables are, (2) how the assumptions of the ANOVA (at least the important ones) were determined to be met, (3) what the results showed (F-statistic, degrees of freedom, exact p value), (4) if any post-hoc tests were conducted and how, etc. Otherwise, this statement is not clear.
>   * The independent variable is modality (i.e. none, equation/table/figure, context, reference). The dependent variable is the performance metric (e.g. rogue score). Since we have a large sample size (n > 30), the Central Limit Theorem (CLT) suggests that the sampling distribution of the mean will be approximately normally distributed, regardless of the shape of the population distribution. We appreciate your feedback and will include the detailed results in the appendix.
>
> * Line 246 appears to hypothesize that referring text for tables summarize the tabulated results, and hence is helpful for question answering. However, if this were the case, wouldn't it mean that the questions can be answered on the basis of the table alone, since the summary can simply be extracted from the table? I.e. the information is already present. Could this be clarified?
>   * You raise an interesting argument, perhaps a better term to use in this situation would be a “natural language description” as opposed to a “summary”.  This description makes higher level reasoning (such as trends) explicit, helping the LLM better answer questions. We will make this clearer in the paper as well.
>
> * What does "authors often utilize tables and figures to supplement reviewer questions" mean exactly? The authors produce the tables and figures before the reviewer questions arise, so how do the authors "supplement" the reviewer questions? I'm not sure I followed the point being made.
>   * Yes, you are correct that the authors produce the tables before hand. However, as evidenced by Table 8, answers refer to tables/equations/figures more often than questions. (In fact this answer serves as a perfect example, where we point the reviewer to Table 8 to answer their question even though their question doesn’t necessarily talk about Table 8).
>
> Some responses to other comments:
>
> * Thank you for pointing us to the PDF extraction resources, perhaps this could be an approach to ensure better matching (though the question of whether using raw LaTeX provides the LLM with better reasoning capabilities remains)
> * We did experiment with question-answer pairs generated by off-the-shelf LLMs, but we found them to be less detailed than those obtained from OpenReview. We have made this data available publicly for the community to use (https://huggingface.co/datasets/avalab/cPAPERS_EQNS_FIGS_GPT). We will add the prompts used to generate this split to the supplementary material so researchers can use this as an additional resource to address the challenges presented in cPAPERS.

---

> > ### Comment · Reviewer_xcYG · 2024-08-16
> >
> > Great, thank you for your detailed response! Since these were my main questions, and I think they’ve been successfully clarified, I’ve raised my score to account for this change.
> >
> > One minor note: I wouldn’t refer to N>30 as a large sample. My question was less about normality (which ANOVA and other parametric methods can be robust to violations of) and more about the other assumptions. In any case, I think a full reporting of results here would be helpful and strongly encourage this.

---

### Official Review · Reviewer_TWp6 · 2024-07-25
**Review of cPAPERS: A Dataset of Situated and Multimodal Interactive Conversations in Scientific Papers**

**Rating:** 7
**Confidence:** 3

**Review:**

The cPAPERS dataset addresses a significant need for multimodal conversational AI systems in scientific research. By grounding question-answer pairs in specific visual and tabular data within papers, it enables more nuanced and contextually relevant interactions. The paper’s methodology for data collection and processing is well-documented, leveraging APIs and regex filtering to ensure accurate extraction of relevant content.

**Strengths:**

Robust Methodology: The detailed and systematic approach to dataset construction ensures high-quality, contextually rich question-answer pairs. Multimodal Integration: The dataset uniquely integrates text, equations, figures, and tables, providing a comprehensive resource for developing conversational AI in scientific literature.
Baseline Approaches: The presentation of baseline approaches using LLMs provides valuable insights and a solid foundation for future research. Relevance and Impact: The work addresses a critical need in the research community, offering tools to manage the growing volume of scientific publications.

**Additional Feedback:**

NA

**Clarity:**

The paper is well-written and organized, making it easy to follow the authors’ approach and understand the significance of the cPAPERS dataset.

**Correctness:**

The paper appears to be technically sound, with a well-defined methodology and appropriate use of tools and techniques for data extraction and processing. The authors utilized APIs and regex filtering to accurately extract relevant content from scientific papers. They also employed a systematic approach for grounding question-answer pairs in specific figures, tables, or equations, ensuring contextual relevance.

**Documentation:**

The dataset is thoroughly documented, with detailed descriptions of the data collection process, filtering steps, and post-processing procedures. This transparency is crucial for reproducibility and future research. The dataset is publicly available at Hugging Face.

**Ethics:**

No.

**Limitations:**

The authors mentioned potential limitations, specifically the presence of mismatched figures, tables, or equations across different versions of manuscripts:

"Limitations. The key limitation of this dataset is the presence of mismatched figures, tables, or equations across different versions of the manuscripts. The .tex files of the paper on arXiv often undergo multiple revisions, and comments on Open Review are typically specific to a particular version. Authors frequently make changes in response to reviewer comments, which may involve additions, removals, or reordering of figures, equations, or tables. This presents a potential mismatch between the figures, equations, or tables and the question-and-answer pairs, thereby introducing additional challenges for language modeling."

While the mentioned limitation is significant, a few other potential limitations could be considered:
Bias in Question-Answer Pairs: The dataset may contain biases based on the specific nature of questions and answers that arise during the peer review process, which could affect the generalizability of models trained on this data.
Domain Specificity: Since the dataset primarily includes papers from NeurIPS and ICLR conferences, it may be biased towards topics prevalent in these venues. Including papers from a broader range of conferences and journals could mitigate this.

**Opportunities For Improvement:**

Enhanced Matching Mechanisms: Implementing more robust techniques to handle mismatches between figures, tables, or equations and the corresponding question-answer pairs due to manuscript revisions could improve accuracy.
Inclusion of More Diverse Paper Types: Expanding the dataset to include a wider variety of scientific papers across different fields could enhance its generalizability and applicability.
Quality Control Measures: Introducing additional quality control measures to verify the accuracy and relevance of the extracted question-answer pairs could further ensure the reliability of the dataset.

**Relation To Prior Work:**

The authors provide a comprehensive review of related work, highlighting the uniqueness of their dataset compared to existing resources. They effectively position cPAPERS within the broader context of multimodal and conversational AI research.

**Summary And Contributions:**

This paper introduces cPAPERS, a novel dataset designed for developing conversational AI systems capable of engaging in multimodal, interactive conversations about scientific papers. The dataset contains 5030 question-answer pairs grounded in figures, tables, and equations from NeurIPS and ICLR papers (2020-2023).
Key contributions include:
Creation of a large-scale dataset for multimodal scientific conversations.
Multimodal Interaction: Supports interactions grounded in visual and tabular data.
Scalable Methodology: Development of a scalable method for data collection and processing.

---

> ### Author Rebuttal · Authors · 2024-08-16
>
> We’d like to thank reviewer TWp6 for their analysis of our paper. We do agree that our dataset is biased towards machine learning questions since we extract pairs from NeurIPS and ICLR. Currently, we are limited by the cost associated with dataset cleanup (through the use of crowd workers) but we will expand on the collection procedure to potentially release a larger v2 of the dataset based on how the community receives this one.

---

### Official Review · Reviewer_yM2s · 2024-07-25
**Good practice to create multi-modal review datasets (Table, Figure, Equation)**

**Rating:** 6
**Confidence:** 4
**Correctness:** See O1-O4.
**Clarity:** In general ok, see O1-O4.

**Review:**

The paper is in general easy to follow, with motivation, clearly stated workflows, and some novelty contributed to the multi-modal academic conversations.

What I missed however is the discussion of dataset usage -- to train conversational agents that do paper digest (like https://www.paperdigest.org/) or to facilitate search of multi-modal material.

**Strengths:**

S1: The paper proposes interesting ideas and approaches to collect conversational pairs via OpenReview articles.

S2: The results about comparing Q, Q+element, Q+ref, Q+context are very interesting and reveal to some extent how authors have created the latex documentation, writing convention and review convention.

**Additional Feedback:**

NA

**Documentation:**

The data are on hugging face (https://huggingface.co/datasets/avalab/cPAPERS/blob/main/README.md) , but with minimal documentation about data set creation, license, usage and stats, etc.

The preprocessing pipeline of datasets and baseline code are not published.

The authors should definitely work on open-sourcing the code to increase the reproducibility of their work.

**Limitations:**

The authors discuss some limitations that in Sec 7.

**Opportunities For Improvement:**

O1: **[Review on similar datasets]**
- LAION datasets like LAION-5B dataset, [1]
- MINT-1T:
Scaling Open-Source Multimodal Data by 10x:A Multimodal Dataset with One Trillion Tokens, [2]


O2: **[Details on data and experimental setup]**
- In Tab 1, how are train/dev/test divided, randomly / stratified by article, or by time?
- Sec 3.1, what is the retrieval rate of papers in NeurIPS & ICLR (2020-2023), how many papers in total? Are the papers all accepted ones or also rejected?
- Sec. 3.2.5: What are the background of these crowdworkers? How to calculate consensus? Sharing of annotation protocols in the manuscript is needed.
- ll. 190-191: As we see later, context (with a paragraph preceding and succeeding the element in the latex source code) could be problematic due to latex formatting constraint. We all know !th sometimes does not work one and one has to manually insert the elements to make them appearing closer to the corresponding text in the compiled pdf.

O3: **[Details on evaluation protocols]**
- ll.241-242: "2x improvement" is not correct. I see the performances are not always two times, Tab 2, yes, Tab 3 no. Make sure to double check the numbers and arguments.
- The authors need to explain in details what the metrics measure. What are the differences between ROUGE-1, ROUGE-2, ROUGE-L, METEOR, BERTScore? There are other measurements like BLEU, why not also use them? See [3]
- ll. 262-264: I do not understand why this makes the Q-based queries, worse.

O4: **[Formatting]**
- The writing styles are sometimes sloppy. Plz double check rigorously. E.g.,
  * Footnote should be after the punctuation, l. 236
  * Space missing, l. 255
  * Title in Sec 3.2.2, double check the formatting of "LATEXfile"
  * Make sure the casing of captions is used consistently, e.g., Tab 1 vs. the remaining. Usually we end a caption with a period (.).
  * I like Fig 1, plz make sure it lies closer to the text describing it. I would suggest the authors add a description below the caption or a summary paragraph in the introduction if the authors want to keep it here.
  * Tab 2-4 should be organized in the same area to facilitate reading.

- I quickly went through the supplementary material (PDF), the authors should work on restructuring (into the main text) and co-reference of important results described in the appendix. Also I would suggest the authors to move the regex description into the appendix.

[1] https://laion.ai/blog/laion-5b/

[2] https://arxiv.org/pdf/2406.11271

[3] https://link.springer.com/article/10.1007/s12046-023-02284-z

**Relation To Prior Work:**

See O1-O3, and [1] - [3].

**Summary And Contributions:**

The authors have presented very interesting datasets based on OpenReview conversations on tables, figures and equations, which are also grounded in ArXiv articles.

They also benchmarked the performance of LLAMA-2-70B in both zero-shot and few-shot settings.

---

> ### Author Rebuttal · Authors · 2024-08-16
>
> We would like to thank reviewer yM2s for their detailed analysis and comments on our paper. Here are some responses to the concerns raised:
>
> O1: Thank you for pointing us to these resources, we will make sure to include them when discussing related work
> O2:
> * In Tab 1, how are train/dev/test divided, randomly / stratified by article, or by time?
>   * Yes, we divide the dataset at random across venues and time
> * Sec 3.1, what is the retrieval rate of papers in NeurIPS & ICLR (2020-2023), how many papers in total? Are the papers all accepted ones or also rejected?
>   * we considered all papers that have been submitted to the main conference. Workshop, special session or tutorials papers are not considered.
>   * We successfully downloaded all submitted papers for ICLR 2020-2023 and NeurIPS 2021 & 2022. However, we were unable to retrieve any papers for NeurIPS 2020 and 2023 due to issues with the OpenReview API.
>   * Some statistics:
>     * ICLR 20: 5688 papers
>     * ICLR 21: 2225 papers
>     * ICLR 22: 8407 papers
>     * ICLR 23: 2773 papers
>     * NeurIPS 21: 9352 papers
>     * NeurIPS 22: 1832 papers
>   * We will add this to the appendix
> * Sec. 3.2.5: What are the background of these crowdworkers? How to calculate consensus? Sharing of annotation protocols in the manuscript is needed.
>   * As described in the supplementary, we recruit crowdworkers with masters qualification from English-speaking countries: Australia, Canada, Ireland, New Zealand, the United Kingdom, and the USA
>   * Line 172 - 174: “For each question-answer pair, two crowdworkers provide feedback on whether the question is technical in nature or not and only those question-answer pairs with a consensus are retained.”
>
> * ll.241-242: "2x improvement" is not correct. I see the performances are not always two times, Tab 2, yes, Tab 3 no. Make sure to double check the numbers and arguments.
>   * Yes, you are correct, which is why we use the phrase “up to 2x improvement” to signify that a 2x improvement is the upper bound
> * The authors need to explain in details what the metrics measure. What are the differences between ROUGE-1, ROUGE-2, ROUGE-L, METEOR, BERTScore? There are other measurements like BLEU, why not also use them? See [3]
>   * Thanks for the suggestion, we will make sure to include them in the paper
>   * ROUGE and METEOR are machine translation metrics that measure n-gram overlap with a reference text while BERTScore measures semantic similarity
>
>   * Thanks for your suggestion, we have computed BLEU scores for all results and we find that they agree with the other trends.
>
> Table 3:
> | Modality       | Setting       | ROUGE-1 | ROUGE-2 | ROUGE-L | METEOR | BERT  | BLEU  |
> |----------------|---------------|---------|---------|---------|--------|-------|-------|
> | Question (Q)   | -             | 0.192   | 0.058   | 0.136   | 0.232  | **0.832** | 0.028 |
> | Q + Table      | Neighboring   | 0.206   | 0.061   | **0.145**   | 0.237  | 0.828 | **0.031** |
> |                | All           | 0.176   | 0.052   | 0.123   | 0.199  | 0.697 | 0.030 |
> | Q + Context    | Neighboring   | 0.202   | 0.062   | 0.142   | 0.241  | 0.828 | **0.031** |
> |                | All           | 0.160   | 0.050   | 0.114   | 0.195  | 0.666 | 0.030 |
> | Q + References | Neighboring   | **0.207**   | **0.064**   | 0.144   | **0.243**  | 0.829 | **0.031** |
> |                | All           | 0.186   | 0.057   | 0.130   | 0.223  | 0.758 | 0.030 |
>
> Table 4:
> | Modality       | ROUGE-1      | ROUGE-2      | ROUGE-L      | METEOR      | BERT        | BLEU        |
> |----------------|--------------|--------------|--------------|-------------|-------------|-------------|
> | Question (Q)   | 0.185        | 0.065        | 0.137        | 0.238       | 0.833   | 0.036       |
> | Q + Caption    | 0.200        | 0.074        | 0.149        | 0.248       | **0.837**   | **0.039**   |
> | Q + Context    | **0.208**    | **0.076**    | **0.155**    | **0.254**   | **0.837**   | **0.039**   |
> | Q + References | 0.205        | 0.075        | 0.154        | 0.251       | **0.837**   | 0.038       |
>
> Table 5:
> | Modality       | ROUGE-1      | ROUGE-2      | ROUGE-L      | METEOR      | BERT        | BLEU        |
> |----------------|--------------|--------------|--------------|-------------|-------------|-------------|
> | Question (Q)   | 0.309        | **0.138**    | 0.248        | 0.215       | **0.860**   | 0.062       |
> | Q + Equation   | **0.317**    | 0.134        | **0.251**    | **0.223**   | **0.861**   | 0.070       |
> | Q + Context    | 0.297        | 0.126        | 0.235        | 0.221       | 0.817       | 0.077       |
> | Q + References | 0.283        | 0.122        | 0.224        | 0.217       | 0.777       | **0.081**   |
>
> Table 6:
> | **Modality**        | **ROUGE-1** | **ROUGE-2** | **ROUGE-L** | **METEOR** | **BERT** | **BLEU** |
> |---------------------|-------------|-------------|-------------|------------|----------|----------|
> | Question (Q)        | **0.315**   | **0.121**   | **0.235**   | 0.218      | **0.869**| 0.059    |
> | Q + Table           | 0.293       | 0.107       | 0.218       | 0.212      | 0.820    | 0.065    |
> | Q + Context         | 0.294       | 0.106       | 0.216       | **0.225**  | 0.838    | **0.070**|
> | Q + References      | 0.292       | 0.106       | 0.214       | 0.218      | 0.816    | 0.068    |
>
> * ll. 262-264: I do not understand why this makes the Q-based queries, worse.
>     * What we mean to say is that authors often refer to equations, tables, or figures in their rebuttals to place their answers in context even though the reviewer’s question did not specifically refer to an equation, table, or figure. As a result, it may not be necessary to utilize information from these modalities to answer some of the reviewer’s questions but providing them distracts the LLM during generation

---

> > ### Author Rebuttal · Authors · 2024-08-16
> >
> > It appears Table 7 was left out, here it is:
> > | **Modality**        | **ROUGE-1** | **ROUGE-2** | **ROUGE-L** | **METEOR** | **BERT** | **BLEU** |
> > |---------------------|-------------|-------------|-------------|------------|----------|----------|
> > | Question (Q)        | **0.329**   | **0.155**   | **0.269**   | **0.250**  | 0.859    | 0.083    |
> > | Q + Caption         | 0.322       | 0.147       | 0.260       | 0.246      | **0.868**| 0.083    |
> > | Q + Context         | 0.321       | 0.140       | 0.256       | **0.250**  | 0.858    | **0.091**|
> > | Q + References      | 0.311       | 0.131       | 0.244       | 0.247      | 0.843    | **0.091**|

---

> > ### Comment · Reviewer_yM2s · 2024-08-31
> > **I'd like to increase my score from 5 to 6 (marginally beyond acceptance threshold).**
> >
> > I'd like to thank the authors for carefully addressing my comments.
> > Most of them are properly addressed and plz make sure you add to the updated version if accepted.
> >
> > However, there are a few points remained unanswered.
> > - O2, point 1: "Yes, we divide the dataset at random across venues and time". Would that have an impact on the results? Do the results depend on the chronological order of train/val/test sets? I would say yes. The authors do not need to run experiments on these chronological splits, but they need to comment on this.
> >
> > - O2, point 2: "However, we were unable to retrieve any papers for NeurIPS 2020 and 2023 due to issues with the OpenReview API." What kind of issues there were with the OpenReview API? For a proper benchmark paper on dataset, the authors would need to document that.
> >
> > - O2, point 5: Thank you for adding the BLEU scores. As expected, the BLEU scores are worse than others yet still maintain the ranking wrt other metrics. I would like to ask the authors to add the explanations to these scores, what are they good for (in model evaluation) and why are they different in performances, how should they be used in benchmark evaluation? These questions are not discussed in the paper. Plz make sure to add them.
> >
> > - In Documentation: I wrote
> > "The data are on hugging face (https://huggingface.co/datasets/avalab/cPAPERS/blob/main/README.md) , but with minimal documentation about data set creation, license, usage and stats, etc. The preprocessing pipeline of datasets and baseline code are not published. The authors should definitely work on open-sourcing the code to increase the reproducibility of their work." These are **not answered** by the authors. I think to make the benchmark more valuable, the authors would need to report to the community about the preprocessing code (e.g., download, preprocessing, ...) and evaluation code (e.g,. zero-shot/fine-tuned LLM input, metrics, ...) on the dataset.

---

> ### Author Response · Authors · 2024-08-30
>
> We would like to ask reviewer yM2s if they had an opportunity to go through our statement and hope that it clarified their concerns. Do let us know if there are any additional details requiring clarification.

---

> ### Author Rebuttal · Authors · 2024-08-31
>
> Thanks for replying to our comments.
>
> - Regarding the comment, "The data are on Hugging Face (https://huggingface.co/datasets/avalab/cPAPERS/blob/main/README.md), but with minimal documentation about dataset creation, license, usage, and stats. The preprocessing pipeline of datasets and baseline code are not published. The authors should definitely work on open-sourcing the code to increase the reproducibility of their work," **we have addressed this concern by publishing the source code for collecting the dataset and reproducing the benchmark results (e.g., zero-shot, fine-tuning, etc.) at [https://github.com/jxu81/cPAPERS](https://github.com/jxu81/cPAPERS).** These have been reported in the [supplementary material](https://openreview.net/attachment?id=DfhcOelEnP&name=supplementary_material) section A.2, as shown below.
>
>   A.2 Dataset Accessibility
>     - The dataset is publicly available at [https://huggingface.co/datasets/avalab/cPAPERS](https://huggingface.co/datasets/avalab/cPAPERS)
>     - URL to Croissant metadata record for viewing and downloading by the reviewers: [https://huggingface.co/datasets/avalab/cPAPERS/blob/main/croissant.json](https://huggingface.co/datasets/avalab/cPAPERS/blob/main/croissant.json)
>     - Code repository for collecting the dataset: [https://github.com/jxu81/cPAPERS](https://github.com/jxu81/cPAPERS)
>     - Code repository for reproducing the benchmark results: [https://github.com/jxu81/cPAPERS](https://github.com/jxu81/cPAPERS)
>
> - Response to Comment O2, Point 2:
>   Regarding the issue with retrieving papers for NeurIPS 2020 and 2023 using the OpenReview API, we encountered a situation where the API returned an empty list of submissions for these years. Notably, the same code successfully retrieved submissions for NeurIPS 2021 and 2022 without any issues. We have been investigating this discrepancy and believe the problem lies with the OpenReview API's endpoint, which is beyond our control. To document this, we will raise the issue in our GitHub repo for future reference and transparency. We will continue to monitor the situation and update our documentation as we gain more insights.
>
>
> **We'd also like to thank you for agreeing to update your assessment, could you please make the change on the official review as well? Our score still shows up as a 5 on the main portal.**

---

### Comment · Area_Chair_mxSW · 2024-08-29
**Reminder to review comments before end of discussion period on 8/31**

Reviewers, thank you for your time and contributions thus far. This is a reminder that the discussion period ends in two days on August 31. Please take some time to engage with the authors' comments and adjust scores if appropriate.

---

### Decision · Program_Chairs · 2024-09-26

**Decision:**

Accept (Poster)

**Comment:**

The paper introduces a useful new dataset for conversational QA about scientific papers and is constructed in a clever way from realistic conversations that happen during the review process. This dataset will be a useful and novel resource for the community. I recommend this paper for acceptance.

**Quality:** Dataset construction methodology is robust. Some additional clarification of statistical methods (ANOVA) and additional baseline comparisons, as discussed by the reviewers, could help strengthen the paper.

**Clarity:** The paper is generally clear and well-organized. Some places for improvement: details of the ANOVA results, and better explanations of how tables and figures relate to reviewer questions.

**Originality:** The authors leverage real-world conversations from the OpenReview platform. This is a creative approach that differentiates cPapers from synthetically constructed alternatives.

**Significance:** The cPAPERS dataset has strong potential to impact the development of conversational AI systems, particularly in the scientific and research domains. The expert-asked and -answered questions and multimodal elements make it a valuable resource. The work could benefit from including a more diverse set of scientific papers.